# Understanding effectiveness of a low-cost food package for ensuring food security during the COVID-19 at the household level: Difference-in-differences analyses of a quasi-experimental trial in Bangladesh

**Md. Golam Rasul[1], Ar-Rafi Khan[1], Md. Ashraful Alam**  **[1], Tahmeed Ahmed[1,2,3], Subhasish Das** [1,4]*

1 Nutrition Research Division, International Centre for Diarrhoeal Disease Research, Dhaka, Bangladesh, 2 Office of the Executive Director, International Centre for Diarrhoeal Disease Research, Dhaka, Bangladesh, 3 Department of Global Health, University of Washington, Seattle, Washington, United States of America, 4 Sydney School of Public Health, University of Sydney, Sydney, Australia

* subhasish.das@sydney.edu.au

## Abstract

### Introduction

The emergence of COVID-19 pandemic compelled to undertake a 'lockdown or shut-down' approach to control the spread of the virus. People from resource-limited settings experienced food insecurity due to lack of supply and access to adequate food during lockdown. Therefore, we developed a low-cost food package and assessed its effectiveness to improve household food insecurity during the COVID-19 pandemic.

### Method

A food package was developed with low-cost, culturally acceptable foods. Each food basket, designed to support a family of four adults and/or one child for 15 days costs 23 USD. A community-based quasi-experimental intervention study was conducted to evaluate the effectiveness of the food package. A total of 245 participants were enrolled in the intervention group, and 244 participants in the control group. A community-based census was carried out to identify vulnerable households and were randomly assigned to the intervention or control groups. Data was collected to assess changes in food insecurity status using the Household Food Insecurity Access Scale, as well as dietary diversity, food frequency, morbidity, and nutritional outcomes.

### Results

Statistically significant changes (p < 0.05) in food insecurity status were observed in the intervention group before, during, and after the intervention. The proportion of food-insecure households decreased from 45.31% to 7.80%, while the proportion of

**Data availability statement:** All data and codebook files are available at: https://dx.doi.org/10.6084/m9.figshare.28642745.

**Funding:** The study was funded by Global Affairs Canada (GAC). The funders had no role in study design, data collection and analysis, decision to publish, or preparation of the manuscript.

**Competing interests:** The authors have declared that no competing interests exist.

food-secure households increased from 8.60% to 31.02% after one month of intervention. The Difference-in-Differences (DID) models estimated a 13.10 percentage point improvement in proportions of food-secure households and a 16.20 percentage point reduction in food-insecure households, both of which were statistically significant (p<0.05).

## Conclusion

A low-cost food intervention improved the food security status of vulnerable families during COVID-19 lockdown. This finding suggests that government and other aid agencies can adopt the developed food package to ameliorate household food insecurity in adverse situations.

## Introduction

On 11th March 2020, the World Health Organization (WHO) declared the SARS- CoV-2 (COVID-19) virus a global pandemic. As of October 2022, 632,560,471 people were affected by COVID-19 worldwide, and more than 6, 581, 817 people died [1]. Due to rapid transmission and prolonged incubation period of the virus, it was not easy to control the spread of COVID-19 [2]. Many countries around the world, including Bangladesh, have had to go for a 'lockdown or shutdown' approach to control the spread of the virus that affected every sphere of life including livelihood, food security, nutrition, social safety nets, food production, and supply chains [3–6]. As a result, people faced various extents of food insecurity across the globe.

Food insecurity refers to a lack of consistent access to enough food for an individual or family caused by limited financial ability or disruption of food availability, access, utilization, and stability [7]. Different studies have identified the extent of food insecurity worldwide due to COVID-19. The World food program reported that due to the impact of the COVID-19 pandemic, 272 million people around the world faced acute food insecurity, and 97 million people suffered from chronic food insecurity [8]. In Britain, after a three-week-long lockdown during the COVID-19 outbreak, about 16.2% of surveyed adults experienced food insecurity, and an additional 21.6% of adults felt worried about availing of the desired food [9]. The study also reported that the percentage of adults experiencing food insecurity was four times higher than before the pandemic [9]. Covid-19 also impacted the livelihood, income, and food security status of the people in the lower- and middle-income countries. In India, a study identified 80% of the surveyed households having less food during the covid-19 lockdown [10]. Another study conducted in six Asia-Pacific countries (India, Myanmar, Vietnam, Bangladesh, Indonesia, and the Philippines) demonstrated that a large number of households had lost their jobs and income and faced different types of food insecurity, including reduced consumption, affordability, and availability of essential food items [11,12]. In Nepal, people faced different types of food insecurity during the COVID-19 period, including unavailability of food, financial constraints for buying food, and price hikes [13,14]. A cross-sectional study conducted in Bangladesh reported that 90% of the rural and urban household suffered from different extent of

food insecurity during the lockdown period [15]. Another study in The Lancet Global Health reported that nearly 70% of rural Bangladeshi households suffered from food insecurity during the COVID-19 lockdown [16]. Globally, studies conducted during the COVID-19 lockdowns, including in Bangladesh, have consistently reported reduced food consumption, lower per capita food expenditure, and decreased household dietary diversity [17–22].

Acute food insecurity during disasters accounts for one-tenth of the global burden. Food baskets are a critical intervention to address this, providing direct support to food-insecure households, supplementary feeding for acutely malnourished groups, and rations for work programs [23]. Previous studies have shown that the food aid programs, particularly during emergencies, significantly enhanced food security for vulnerable populations. However, challenges in coverage and sustainability persist [24,25]. Therefore, this study aimed to develop a reliable food package to alleviate food insecurity and mitigate the negative impacts of COVID-19 and similar public health emergencies during disasters.

## Methods

### Study site

We conducted a community-based quasi-experimental intervention trial among the residents of one urban (Bauniabadh and adjacent slum area, Mirpur, Dhaka) and one rural area (Matlab, Chandpur) of Bangladesh. Mirpur has a population of more than half a million in an area of 14.22 km$^2$. More than 38, 000 people live in each square kilometer of the area compared with the mean of 8229/km$^2$ in Dhaka district and 976/km$^2$ in Bangladesh. Most people are day laborers, garment workers, and transport workers [26]. The rural field site, Matlab, is in the Chandpur district, about 55 km southeast of Dhaka. Since 1966, icddr,b has been running an internationally recognized and unique Health and Demographic Surveillance System (HDSS) involving all 142 villages in the upazila (sub-district) comprising a population of 230,000. In Matlab, most people are agricultural workers, day laborers, and fishermen [27].

### Selection of the study participants

Participants of the intervention and control group were carefully selected from households (HH) with similar socio-economic conditions. A community-based census was conducted in the study areas to collect the necessary background information. Participants who had missed any meal during last month due to food shortage, had lost their livelihood due to the pandemic, had a monthly income less than or equal to 15000 BDT and were willing to consent to participate in the study were included. HHs that already received any food aid during the household listing were excluded from the study. After the household listing, a sampling frame was created, and participants were randomly assigned to either the intervention or control group. A Field Research Assistant (FRA) approached eligible families, explained the study in language appropriate to their literacy level, and obtained written informed consent from the HH head. The study was approved by the Institutional Review Board of the International Centre for Diarrhoeal Disease Research, Bangladesh (protocol no: 20120).

### Intervention

A survival food pack was developed with low-cost and culturally accepted food, including grains, legumes and nuts, lentils, flesh food, vitamin-A-rich fruits and vegetables, other vegetables, oil, salt, sugar, dry foods, and vitamin – C-rich food. Nutritional values were calculated using a standard food composition table [28], and the amount of food consumption per person was calculated using the dietary guidelines for Bangladesh [29]. This guideline mentioned the desirable dietary intake of adults and children based on different research. Also, the world health organization (WHO) recommended energy requirement for an adult person is 2420 kcal per day, and for a child is 1290 kcal [30]. A technical expert group of the Government of Bangladesh also provided a technical report titled "Report on Food Packages for Disaster Affected Population" in 2020. This report suggested ensuring at least 2121 kcal for an adult person and 1094 kcal energy for a child in a food basket during any disasters. This report also indicated that this food package should contain food for at

least ten days, be locally available, culturally acceptable, low cost, and fulfill the nutrition and dietary requirement of the affected population [31]. Considering all the issues, we developed a food basket (Table 1) that contains 2443 kcal for an adult person and 1145 kcal for a child.

In our study, we delivered soyabean chunk as an alternative to flesh food. It is cost-effective, easy to distribute, easy to store, and contains good amounts of energy and protein. This food package was distributed to each family for three months. A 15-days delivery basket for a family of four adult members and one child costs 24 USD (Table 2). In addition, both the groups received Behavioral Change Communication messages to prevent COVID-19. However, the control group did not receive any direct food package or dietary intervention.

## Sample size

We assumed that 60% of the households might suffer from severe food insecurity during the pandemic period and the prevalence would go down to 40% upon utilizing the food package. Hence, considering an effect size of 20% and a 5% chance of type 1 error, at 20% power, the minimum sample size was 98 in each group. With a chance of 20% dropout during the study period, the number of families we needed to recruit in each group per site was 122. Finally, a total of 489 households were enrolled in our study. Among them, 245 households (121 urban and 124 rural) were enrolled in the intervention group, and 244 households (121 urban and 123 rural) were enrolled in the control group.

## Data collection

Data were collected three times from both groups to understand the changes in food insecurity, dietary diversity, morbidity, and nutritional status of the household members during the study period. Baseline data were collected before the intervention started, during intervention data were collected after one month of the intervention started and end-line data were collected one month after the intervention completed. For household food insecurity data, female adults were interviewed using the Household Food Insecurity Access Scale (HFIAS) guideline [32]. The HFIAS documents the experience of food insecurity based on the lack of access due to poverty rather than a supply shortage and captures the predictable reactions

Table 1. Standard food package for the participants.

| Serial | Food groups | Specific food item to be offered | Amount/day (for one adult member) | Amount/day (for one child) | Energy/100 gm (kcals) | Energy/day/unit of food) Adult | Energy/day/unit of food) Child |
|---|---|---|---|---|---|---|---|
| 1 | Grains, roots and tubers | Rice | 345 | 75 | 346 | 1194 | 260 |
| 2 | Legumes and nuts | Pulse | 36 | 20 | 317 | 114 | 63 |
| 3 | Lentils | Motor dal | 40 | | 326 | 130 | |
| 5 | Flesh foods (meat, fish, poultry and liver/organ meats) | Soybean | 20 | | 424 | 85 | 0 |
| 6 | Eggs | Egg | 1 | 0.5 | 158 | 158 | 79 |
| 7 | Vitamin-A rich fruits and vegetables | Pumpkin | 40 | | 29 | 11.6 | |
| 8 | Other fruits and vegetables. | Potato | 100 | 20 | 66 | 66 | 33 |
| 9 | Fortified vegetable oil | Fortified vegetable oil | 40 | 30 | 900 | 360 | 180 |
| 10 | | Iodised salt | | | 0 | 0 | 0 |
| 11 | | Sugar/ Molasses | 50 | 25 | 398 | 199 | 99.5 |
| 12 | | Suji | 36 | 50 | 346 | 125 | 173 |
| 13 | | Biscuits | | 75 | 344 | 0 | 258 |
| 14 | Vitamin- C | Lemon | 0.2 | | 56 | 0.11 | |
| | Total | | | | | 2443 | 1145 |

**Table 2. Cost of food packages for 15 days.**

| Serial | Food groups | Specific food items offered | 15 days/ adult | 15 days child | Total for a family | Rounded Amount | Unit Price/ kg/ pcs* | Total Price (BDT) | Total Price USD |
|---|---|---|---|---|---|---|---|---|---|
| 1 | Grains, roots and tubers | Rice | 13.8 | 1.1 | 14.92 | 15 | 45 | 675 | 7.94 |
| 2 | Legumes and nuts | Pulse | 1.4 | .300 | 1.74 | 1.5 | 85 | 127.5 | 1.50 |
| 3 | Lentils | Motor dal | 1.6 | 0 | 1.6 | 1.5 | 70 | 105 | 1.23 |
| 5 | Flesh foods | Soybean nuggets | 0.80 | 0 | .800 | 0.8 | 200 | 160 | 1.88 |
| 6 | Eggs (pcs) | Egg | 40 | 8 | 48 | 45 | 8 | 360 | 4.23 |
| 7 | Vitamin-A rich fruits and vegetables | Pumpkin | 1.6 | 0 | 1.6 | 1.6 | 30 | 48 | 0.56 |
| 8 | Other fruits and vegetables. | Potato | 4.0 | 0 | 4.0 | 4 | 20 | 80 | 0.94 |
| 9 | Fortified vegetable oil | Fortified vegetable oil | 1.60 | 0.300 | 1900 | 1.8 | 100 | 180 | 2.11 |
| 10 | | Iodized salt | 0 | 0 | 0 | 0.5 | 30 | 15 | 0.017 |
| 11 | | Sugar/ Molasses | 1.0 | 0.750 | 1750 | 1.5 | 60 | 90 | 1.05 |
| 12 | | Suji | 1.4 | 0.750 | 2190 | 2 | 60 | 120 | 1.41 |
| 13 | | Biscuits | 0 | 0.750 | 750 | 0.5 | 120 | 60 | 0.70 |
| 14 | Vitamin- C | Lemon (pcs) | 8 | 0 | 8 | 8 | 3 | 24 | 0.28 |
| | | | | | | | | | 23.84 |

and responses. It summarizes the food insecurity on a scale of 1–3 [32,33]. For anthropometry data, all the measurements (height, weight, MUAC) were taken by two trained FRAs following standard operating procedures and using standard tools. The SECA baby scales and infantometer were used for measuring weight and length. To measure dietary diversity at the household and personal level, Household Dietary Diversity Score (HDDS) and Food Frequency Questionnaires (FFQ) were used, respectively. To measure the minimum dietary diversity score of the children from 6–24-month age, FFQ data were collected from the caregiver of the child in each household.

## Outcome variables

The primary outcome variable was the changes in household food security status. The secondary outcomes were the changes in the anthropometric status (LAZ, WAZ, WLZ scores, MUAC for the children of 6–59-months age and BMI for the adults), and the dietary diversity status of the household members.

## Data analysis

We reported the overall socio-economic status of the study participants including sex and occupation of household head, education, asset index, income and age, household characteristics, and food security status using mean, standard deviation (SD), frequency, and percentages. Household Food Insecurity Access Prevalence (HFIAP) was calculated from the HFIAS scores, which reflect a household's level of access to food [34]. The households were classified as food secure, mild, moderate, and severe food insecure. Mild and moderate food insecure households were combined into one category. Chi-square test or Fisher's exact test was performed to compare the household-differences in categorical variables.

The difference-in-differences (DID) analysis, a quasi-experimental data analysis technique, was used to measure the effectiveness of the delivered intervention on food security status of the household. In the regression models, a probability value or p-value less than 0.05 was considered statistically significant, and regression coefficients with 95% confidence intervals (95% CIs) were reported. The multivariable DID models were adjusted for different socio-economic

characteristics of the participants. These variables were- having under-five children in the household, gender of household head, their occupation, age, income status of the housewives, their age, education, household's accessibility to financial services if the household had any agricultural land, chicken-duck, their asset index and monthly income.

DID modeling technique measures the actual effect of the intervention on the outcome and its before-after changes with the following formula: DID: [(B − A) − (D − C)], where

A = Baseline prevalence of food insecurity in the intervention group,

B = End-line prevalence of food insecurity in the intervention group,

C = Baseline prevalence of food insecurity in the comparison group, and

D = End-line prevalence of food insecurity in the comparison group.

To assess the true effect of the intervention, we will use a regression model with a generalized estimating equation as follows: Yit = β0 + β1Time + β2Group + δ (Time × Group) + β3X + ε [2] where,

Yit = outcome variable of interest for individual i at time t,

Time = [1] if end-line and (0) if baseline,

Group = [1] if intervention group and (0) if comparison group,

δ = the effect of nutritional intervention, X = other covariates, and ε = error term [35].

The effect of the intervention on the other outcome variables was analyzed following the same technique.

## Results

Among the study participants, most of the household were male headed, and the majority of the household head were day laborers. In a comparison between the intervention and control groups, participation in income-generating activities among housewives was marginally higher in the intervention group. The distribution across economic quintiles showed a similar trend for both the groups, with minor variations in the middle and richest categories. The average ages of the household heads and housewives were comparable across both the groups. However, the average monthly household income was somewhat higher in the control group than in the intervention group. The detailed socio-demographic characteristics of the study participants are presented in Table 3.

Figs 1, 2, and Table 4 present the changes in household food insecurity levels over time for both the intervention and control groups. At the baseline, severe food insecurity was significantly higher in the intervention group, whereas mild-to-moderate food insecurity was more prevalent in the control group. After the first month of intervention, the intervention group experienced substantial improvements, including a significant decrease in both severe and mild-to-moderate food insecurity, along with a significant increase in the number of food-secure households. In contrast, the control group experienced a modest change, with only a slight reduction in severe food insecurity and a less pronounced overall shift. At the endline, both groups had improved food security levels, but the intervention group achieved higher levels of food security compared to the control group. While some instances of severe food insecurity persisted in both the groups, mild-to-moderate food insecurity continued to be more common in the control group.

Also, Table 4 present the changes in the household dietary diversity and food consumption patterns of children aged 6–24 months over the course of the study. Both the intervention and control groups showed improvements in household dietary diversity scores, although the control group experienced a slightly higher increase by the end of the study. For children's dietary diversity, the control group demonstrated a higher increase in minimum dietary diversity scores over time than the intervention group. Additionally, study results indicated that, at baseline, most households reduced their food consumption, borrowed money or food, and altered their primary food sources as coping strategies. These practices remained consistent among the control group throughout the intervention and after the intervention completed. In contrast, the intervention group showed a significant reliance on food support from the project at the midline. By the endline, many households reported changing their food habits and borrowing money to purchase food as coping mechanisms.

**Table 3. Background Information of the study participants.**

| | Intervention | Control | p-value |
|---|---|---|---|
| | N (%) | | |
| | | | |
| Food secure | 21 (8.6) | 34 (13.9) | < 0.01 |
| Mild-to-moderate food insecure | 113 (46.1) | 141 (57.8) | |
| Severely food insecure | 111 (45.3) | 69 (28.2) | |
| Having under 5 children in the household | | | |
| Yes | 151 (57.6) | 156 (63.9) | 0.15 |
| Head of the household | | | |
| Male | 207 (84.5) | 223 (91.4) | 0.02 |
| Occupation of the household head | | | |
| Not currently working | 34 (13.9) | 26 (10.7) | 0.01 |
| Fishing/ agricultural worker | 5 (2.0) | 17 (6.9) | |
| Day labor | 50 (20.4) | 51 (20.9) | |
| Businessman | 41 (16.7) | 55 (22.5) | |
| Professional/ wage earner | 57 (23.3) | 59 (24.2) | |
| Other | 58 (23.7) | 36 (14.8) | |
| Housewives involved in earning | | | |
| Yes | 31 (12.7) | 28 (11.5) | 0.67 |
| Have bank account | | | |
| Yes | 64 (26.1) | 55 (22.5) | 0.37 |
| Owns chicken or ducks | | | |
| Yes | 74 (30.2) | 62 (25.4) | 0.29 |
| household own any agricultural land | | | |
| Yes | 15 (6.1) | 20 (8.2) | 0.37 |
| Asset Index | | | |
| Poorest | 97 (39.6) | 99 (40.6) | 0.70 |
| Middle | 51 (20.8) | 47 (19.3) | |
| Richer | 44 (18.0) | 54 (22.1) | |
| Richest | 53 (21.6) | 44 (18.0) | |
| Residence | | | |
| Urban | 121 (49.4) | 121 (49.6) | 0.96 |
| Rural | 124 (50.6) | 123 (50.4) | |
| Mean (SD) | | | |
| Education of respondents (in years) | | | |
| HH Head | 4.2 (3.8) | 4.3 (3.9) | 0.56 |
| Mother/Housewife | 3.8 (3.4) | 3.3 (3.5) | 0.24 |
| Monthly income of the HH ($) | 95.7 (38.9) | 105.2 (46.6) | 0.62 |
| Age of the household head | 42.8 (11.2) | 43.2 (11.8) | 0.98 |
| Age of the mother/housewife | 35 (10.2) | 35.3 (11) | 0.89 |

The DID regression results indicated the impact of the intervention on food security status, household dietary diversity, and food consumption among children aged 6–24 months in both the intervention and control groups. After adjusting all covariates, the intervention group showed a reduction in severe food insecurity, with a significant p-value indicating improvement. Additionally, the food security status among the intervention households increased significantly. However,

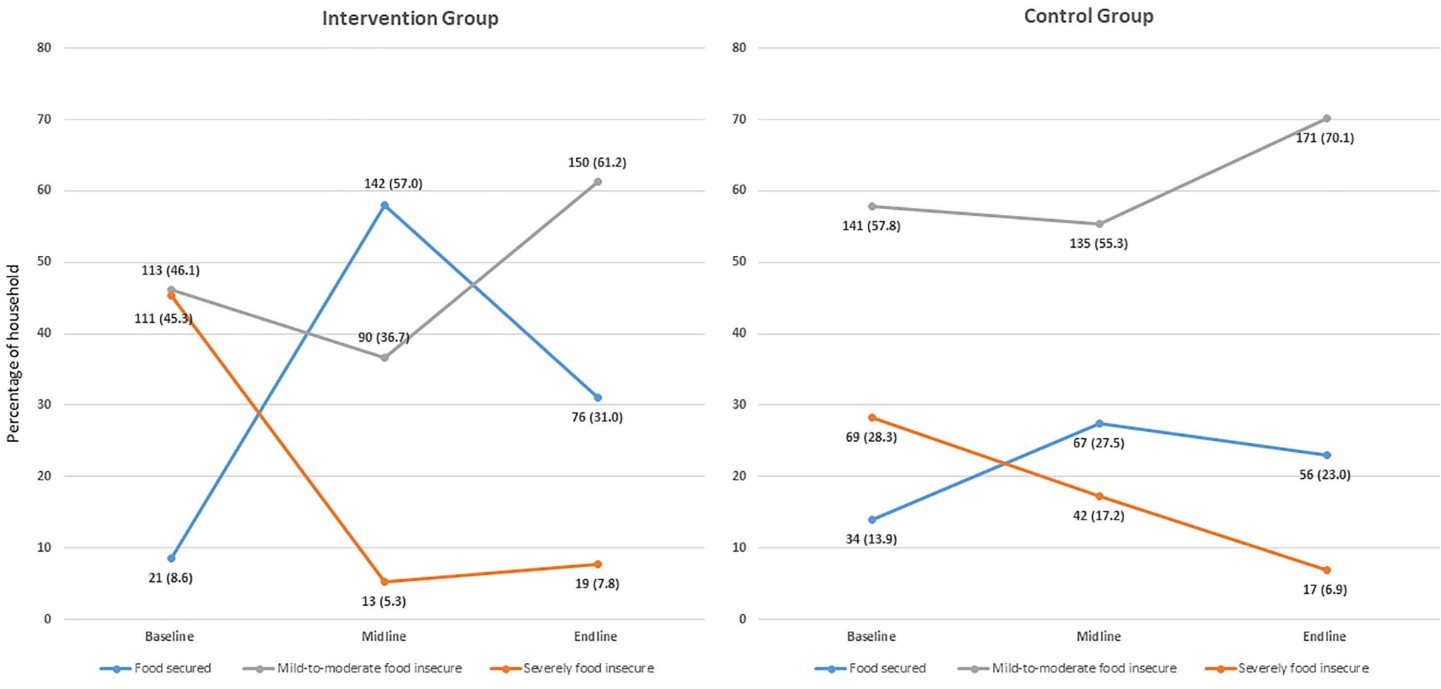

**Fig 1. Overall changes in household food security status in different stages of intervention.**

the change in the household dietary diversity score was not statistically significantly different between the groups, suggesting that the intervention had a minimal impact on dietary diversity when comparing baseline and end-line data.

## Discussion

This study aimed to assess the household food insecurity, nutritional status, and dietary diversity during the COVID-19 pandemic. Additionally, it sought to design an effective food package that could alleviate food insecurity and mitigate the adverse effects of the COVID-19 pandemic, as well as similar epidemics, pandemics, and emergency or disaster situations. Our study results showed that the food insecurity status of the household decreased drastically during the intervention period as the family were receiving food package. However, the efficiency of food aid as both the short-term and long-term coping strategy is still questionable [36]. Food aid procedure has been criticized for lack of timelines, high food delivery cost, inefficiency of supply chain management, and high administrative cost [37,38]. Apart from these drawbacks, shot-term food aid can reduce the food price in society, increase producer and receiver's income, and help the community to reduce long-term food insecurity [36]. On the other hand, our study results showed that food insecurity status also improved for households who didn't receive any food assistance simultaneously. After one month of the intervention, both groups' food insecurity status was improved compared to the baseline prevalence. It indicates that short-term food intervention positively impacts the long-term food security status of the receiving household and society.

Previous studies also mentioned that most households rely on food aid or relief, decreasing food consumption and skipping meals, taking loans or selling assets to meet their daily food needs during any disaster [15,39,40]. Different studies have mentioned the importance of food aid during this pandemic or any other disaster when regular food supplies is interrupted or people can't afford to purchase food [41,42]. A previous study in the same setting also illustrated similar coping strategies. Though some households received different types of food and financial support from various sources, no systematic relief or safety net program was implemented in those areas during the pandemic [15]. In our study we

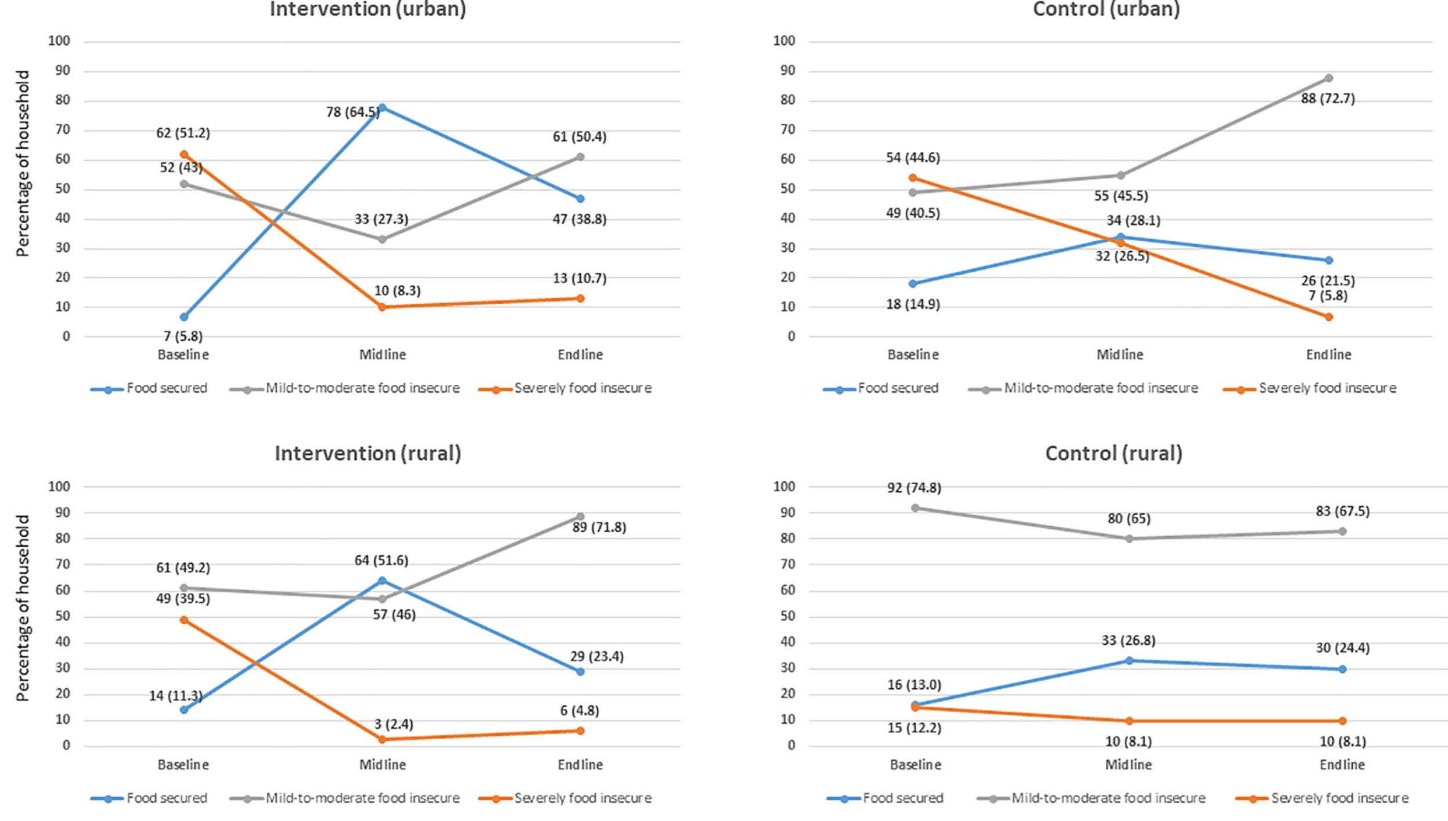

**Fig 2. Group specific household food security status in different stages of intervention.**

enrolled study participants based on specific criteria and provided food aid for three months. Also, adequate food consumption, nutrition value, price, availability, and acceptance of the specific food were considered in the design of the food package. This strategy of food aid as a short-term coping strategy played an important role in improving the household's short-term and long-term food security.

Different studies have mentioned numerous constraints for successfully implementing food aid programs. Proper supply chain management, procurement of food, coordination among various stakeholders, selection of households at risk of food insecurity, and need assessment of the intervention household are connected to a successful implementation of a food distribution program [19,43–45]. A study in Bangladesh pictured a comparison of supply chain management between Bangladesh and developed countries. The developed countries maintain their humanitarian supply chain system in a need-based pull system and proper estimation and distribution of their relief system whereas in Bangladesh, the supply system runs on a central forecasting system [46–48]. This study recommended considering the local people's needs, coordinating with different stakeholders, and preparing a list of standardized food items to be delivered during any crisis moment [18]. In our study, we examined the effectiveness of standardized food baskets considering the needs of the local people, acceptability, and availability of food items which meet the population's nutritional requirements. To maintain the smooth supply chain system, we collaborated with the local supplier, prepared a total forecast, and shared it with the supplier to ensure a smooth supply of consumable items.

We did not provide any direct intervention to the control group. But their socio-economic conditions could potentially confound the findings of our study. Statistical modeling using DID technique had adjusted the effect for such confounding. Hence, we believe we successfully captured our participants' true changes in food security status.

**Table 4. DID estimates of the effects of the intervention on household food security and nutritional status of the study population.**

| | | | Control | | | Intervention | | | DID | | |
|---|---|---|---|---|---|---|---|---|---|---|---|
| | | | N | % (SD) | 95% CI | N | % (SD) | 95% CI | Coefficient | 95% CI | p-value |
| Household food security status | Secure | Baseline | 34 | 13.93 (2.22) | 9.58, 18.29 | 21 | 8.57 (1.79) | 5.06, 12.08 | | | |
| | | Endline | 56 | 22.95 (2.69) | 17.67, 28.23 | 76 | 31.02 (2.96) | 25.22, 36.82 | 13.13 | 3.9, 22.35 | <0.01 |
| | Mild to modarate | Baseline | 141 | 57.79 (3.16) | 51.58, 63.99 | 113 | 46.12 (3.18) | 39.87, 52.37 | | | |
| | | Endline | 171 | 70.08 (2.93) | 64.33, 75.83 | 150 | 61.22 (3.11) | 55.12, 67.33 | 3.26 | −8.7, 15.23 | 0.59 |
| | Severely insecure | Baseline | 69 | 28.28 (2.88) | 22.62, 33.94 | 111 | 45.31 (3.18) | 39.07, 51.55 | | | |
| | | Endline | 17 | 6.97 (1.63) | 3.77, 10.17 | 19 | 7.76 (1.71) | 4.4, 11.11 | −16.39 | −25.66, −7.12 | <0.01 |
| Age 6–24 month | Minimum dietary diversity | Baseline | 14 | 8.33 (2.13) | 4.15, 12.52 | 13 | 8.23 (2.19) | 3.94, 12.52 | | | |
| | | Endline | 17 | 10.49 (2.41) | 5.77, 15.22 | 15 | 9.87 (2.42) | 5.12, 14.62 | −0.12 | −9.13, 8.9 | 0.98 |
| | Minimum meal frequency | Baseline | 27 | 16.07 (2.83) | 10.51, 21.64 | 35 | 22.15 (3.3) | 15.66, 28.64 | | | |
| | | Endline | 28 | 17.28 (2.97) | 11.45, 23.12 | 25 | 16.45 (3.01) | 10.54, 22.35 | −5.49 | −17.39, 6.42 | 0.37 |
| | Minimum acceptable diet | Baseline | 12 | 7.14 (1.99) | 3.24, 11.04 | 11 | 6.96 (2.02) | 2.99, 10.94 | | | |
| | | Endline | 14 | 8.64 (2.21) | 4.31, 12.98 | 12 | 7.89 (2.19) | 3.6, 12.19 | −0.07 | −8.37, 8.24 | 0.99 |
| | | | N | mean (SD) | 95% CI | N | mean (SD) | 95% CI | Coefficient | 95% CI | P-value |
| Age 6–59 month | Length for Age Z score | Baseline | 168 | −1.48 (0.1) | −1.67, −1.29 | 158 | −1.46 (0.09) | −1.64, −1.28 | | | |
| | | Endline | 160 | −1.51 (0.09) | −1.69, −1.34 | 152 | −1.45 (0.08) | −1.61, −1.29 | 0.05 | −0.29, 0.39 | 0.77 |
| | Weight for height Z score | Baseline | 168 | −0.36 (0.09) | −0.54, −0.18 | 158 | −0.48 (0.08) | −0.64, −0.31 | | | |
| | | Endline | 160 | −0.3 (0.1) | −0.5, −0.1 | 152 | −0.61 (0.08) | −0.77, −0.45 | −0.19 | −0.55, 0.16 | 0.29 |
| | Weight for age Z score | Baseline | 168 | −1.09 (0.09) | −1.27, −0.91 | 158 | −1.15 (0.08) | −1.31, −0.99 | | | |
| | | Endline | 160 | −1.06 (0.1) | −1.26, −0.86 | 152 | −1.23 (0.08) | −1.39, −1.07 | −0.11 | −0.45, 0.24 | 0.55 |
| Age 10–19 year | BMI for age Z score | Baseline | 169 | −0.57 (0.12) | −0.8, −0.34 | 186 | −0.44 (0.11) | −0.65, −0.23 | | | |
| | | Endline | 173 | −0.6 (0.12) | −0.83, −0.37 | 196 | −0.53 (0.1) | −0.72, −0.33 | −0.11 | −0.54, 0.33 | 0.64 |
| Age > 19 year | Calculated BMI | Baseline | 634 | 23.8 (0.19) | 23.43, 24.18 | 677 | 23.45 (0.17) | 23.12, 23.78 | | | |
| | | Endline | 641 | 23.5 (0.19) | 23.12, 23.87 | 684 | 23.51 (0.17) | 23.17, 23.85 | 0.31 | −0.4, 1.02 | 0.39 |
| HH dietary diversity | HDD Score | Baseline | 244 | 7.68 (0.1) | 7.49, 7.88 | 245 | 7.64 (0.1) | 7.45, 7.84 | | | |
| | | Endline | 244 | 8.62 (0.1) | 8.41, 8.82 | 245 | 8.38 (0.1) | 8.18, 8.58 | −0.34 | −0.7, 0.01 | 0.06 |

## Conclusion

This study demonstrated that short-term food aid significantly alleviated the household food insecurity during the COVID-19 pandemic, with positive effects extending beyond the intervention period. While food aid played a critical role in emergencies, challenge such as logistical inefficiencies remained. Future efforts should focus on refining supply chain systems and integrating behavioral interventions to ensure sustainable food security solutions during crises.

## Limitation and future research direction

Like any experimental study, this research has limitations that could be addressed through future research. In this study, we provided intervention to a limited number of people in two specific areas due to budget constraints. However, distributing food baskets in diverse areas could face different problems that could not be identified. Also, we selected the study participant from comparable socio-economic backgrounds and did not explicitly investigate the potential influence of pre-existing socio-economic differences on the outcomes, presenting a notable limitation. Another limitation of this study was the baseline imbalance in food security status between intervention and control groups. The absence of initial matching could affect comparability. Future studies should consider baseline matching of food insecurity status to improve

the group-wise equivalency and strengthen causal interpretations of the intervention effects. Additionally, in this study, we purchased the food item in the most convenient procedure, while the government follows different supply chain and procurement procedures. But the insights from the government officials and other related stakeholder were not incorporated in this study. Hence further research is needed to understand the acceptability of integrating this food package into large-scale safety net programs implemented by the government and other stakeholders.

## Acknowledgments

The authors would like to thank all the participants for sharing their time and providing consent and information necessary for the successful completion of the study. The authors also acknowledge the contribution of icddr,b's core donors including Government of the People's Republic of Bangladesh and Global Affairs Canada (GAC), Canada; for their continuous support and commitment to the icddr,b's research efforts.

## Author contributions

**Conceptualization:** Md. Golam Rasul, Tahmeed Ahmed, Subhasish Das.

**Data curation:** Md. Golam Rasul, Ar-Rafi Khan, Md Ashraful Alam, Subhasish Das.

**Formal analysis:** Md. Golam Rasul, Ar-Rafi Khan, Md Ashraful Alam.

**Investigation:** Subhasish Das.

**Methodology:** Subhasish Das.

**Project administration:** Subhasish Das.

**Resources:** Subhasish Das.

**Supervision:** Md. Golam Rasul, Tahmeed Ahmed, Subhasish Das.

**Writing – original draft:** Md. Golam Rasul, Subhasish Das.

**Writing – review & editing:** Md. Golam Rasul, Ar-Rafi Khan, Md Ashraful Alam, Tahmeed Ahmed, Subhasish Das.

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
