## [Decision Letter · Decision Letter 0]

29 Nov 2023

PONE-D-22-31934Developing and assessing the effectiveness of a low-cost food package for ensuring food security at the household level during pandemic: A community-based quasi-experimental trialPLOS ONE

Dear Dr. Das,

Thank you for submitting your manuscript to PLOS ONE. After careful consideration, we feel that it has merit but does not fully meet PLOS ONE’s publication criteria as it currently stands. Therefore, we invite you to submit a revised version of the manuscript that addresses the points raised during the review process.

We look forward to receiving your revised manuscript.

Kind regards,

Noor Raihani Zainol

Academic Editor

PLOS ONE

Journal Requirements:

Additional Editor Comments (if provided):

Reviewer 1

Thanks very much for the opportunity to review this interesting article. The development of a low-cost food basket is a much needed intervention in the face of COVID-19 and other global threats, and evaluation of such an effort is critical. While factors contributing to the creation of the intervention is shared, more detail as to the targeting and overall approach would be valuable. The methodological details is very strong in some areas, but omits key information. For example, there is not a clear picture of how control and intervention groups were sorted. This is particularly glaring because they are unbalanced in the outcome variable of interest at baseline. It’s possible that differences in findings thus are attributable to their initial differences in starting points, as opposed to the intervention. This point is not discussed, and should be identified in the text as a key shortcoming of this study.

Additional comments are included as follows:

Check presentation of the findings in the abstract carefully, will be important to address methods used more clearly.

Check for spelling errors in the abstract (eg, improv for impove)

The writing throughout is organized into very long paragraphs. It would be more digestible if the paragraphs were more targeted.

Introduction: There has been a lot of food insecurity data about the pandemic available at this point; studies in Britain are not the most relevant to the Bangladesh context. More global and Bangladeshi (if available) studies would be better context.

Orienting the introduction on acute food insecurity in disasters and the ways such a food package might have broader relevance to other scenarios (even beyond epidemic/pandemic) would be a valuable approach.

4 adults and 1 child is an odd target for the intervention given that’s not likely the average structure of households. How and why was this selected?

This was paired with extensive outcome measures for children, but if the intervention is only targeted to one child in the household how was this addressed?

The study design is not adequately described. How were households determined to be intervention or control households? Was this randomized?

Selection criteria of food insecurity is included, but then households that are food secure are in the sample.

Substantial discussion of the uneven sample is needed – far more individuals were in the food insecure category for the intervention group, allowing for a great potential impact if the intervention is most effective for the severely food insecure. The limitations fail to discuss this.

Reviewer 2

Editing is necessary.

Abstract: More information on selecting intervention and control group must be presented.

Introduction: There are no literature on the effectiveness of food baskets to improve food insecurity in Bangladesh or other countries.

Methods: Considerations for developing food package needs references.

Which criteria was used to select the control group?

How many months did the intervention last?

We used the Household Food Insecurity Access Scale to calculate the Household Food Insecurity Access Prevalence (HFIAP) based on their responses on the level of access to food and according to their score (Ref???).

Repeated measure ANOVA is recommended to detect the differences between intervention and control group adjusted for other variables.

Results: The text of the findings is a repetition of the numbers in Tables and figures. Only significant differences is important to mention.

Figures 1 and 2 were not accessible.

Discussion: Some discussions are unrelated to the results:

“We found similar coping strategies in urban and rural areas in the baseline data. Some households received different types of support from various sources, but no systematic relief or safety net program was implemented in those areas.”

Comparisons of effectiveness with other studies have not been done.

Acknowledgement: The authors would like to thank all the participants and their parents…???

Reviewers' comments:

Reviewer's Responses to Questions

**Comments to the Author**

1. Is the manuscript technically sound, and do the data support the conclusions?

Reviewer #1: No

Reviewer #2: Partly

2. Has the statistical analysis been performed appropriately and rigorously? 

Reviewer #1: I Don't Know

Reviewer #2: No

3. Have the authors made all data underlying the findings in their manuscript fully available?

Reviewer #1: Yes

Reviewer #2: Yes

4. Is the manuscript presented in an intelligible fashion and written in standard English?

Reviewer #1: Yes

Reviewer #2: No

5. Review Comments to the Author

Reviewer #1: Thanks very much for the opportunity to review this interesting article. The development of a low-cost food basket is a much needed intervention in the face of COVID-19 and other global threats, and evaluation of such an effort is critical.

While factors contributing to the creation of the intervention is shared, more detail as to the targeting and overall approach would be valuable.

The methodological details is very strong in some areas, but omits key information. For example, there is not a clear picture of how control and intervention groups were sorted. This is particularly glaring because they are unbalanced in the outcome variable of interest at baseline. It’s possible that differences in findings thus are attributable to their initial differences in starting points, as opposed to the intervention. This point is not discussed, and should be identified in the text as a key shortcoming of this study.

Additional comments are included as follows:

Check presentation of the findings in the abstract carefully, will be important to address methods used more clearly.

Check for spelling errors in the abstract (eg, improv for impove)

The writing throughout is organized into very long paragraphs. It would be more digestible if the paragraphs were more targeted.

Introduction: There has been a lot of food insecurity data about the pandemic available at this point; studies in Britain are not the most relevant to the Bangladesh context. More global and Bangladeshi (if available) studies would be better context.

Orienting the introduction on acute food insecurity in disasters and the ways such a food package might have broader relevance to other scenarios (even beyond epidemic/pandemic) would be a valuable approach.

4 adults and 1 child is an odd target for the intervention given that’s not likely the average structure of households. How and why was this selected?

This was paired with extensive outcome measures for children, but if the intervention is only targeted to one child in the household how was this addressed?

The study design is not adequately described. How were households determined to be intervention or control households? Was this randomized?

Selection criteria of food insecurity is included, but then households that are food secure are in the sample.

Substantial discussion of the uneven sample is needed – far more individuals were in the food insecure category for the intervention group, allowing for a great potential impact if the intervention is most effective for the severely food insecure. The limitations fail to discuss this.

Reviewer #2: Editing is necessary.

Abstract: More information on selecting intervention and control group must be presented.

Introduction: There are no literature on the effectiveness of food baskets to improve food insecurity in Bangladesh or other countries.

Methods: Considerations for developing food package needs references.

Which criteria was used to select the control group?

How many months did the intervention last?

We used the Household Food Insecurity Access Scale to calculate the Household Food Insecurity Access Prevalence (HFIAP) based on their responses on the level of access to food and according to their score (Ref???).

Repeated measure ANOVA is recommended to detect the differences between intervention and control group adjusted for other variables.

Results: The text of the findings is a repetition of the numbers in Tables and figures. Only significant differences is important to mention.

Figures 1 and 2 were not accessible.

Discussion: Some discussions are unrelated to the results:

“We found similar coping strategies in urban and rural areas in the baseline data. Some households received different types of support from various sources, but no systematic relief or safety net program was implemented in those areas.”

Comparisons of effectiveness with other studies have not been done.

Acknowledgement: The authors would like to thank all the participants and their parents…???

6. PLOS authors have the option to publish the peer review history of their article (what does this mean? ). If published, this will include your full peer review and any attached files.

**Do you want your identity to be public for this peer review?** For information about this choice, including consent withdrawal, please see our Privacy Policy .

Reviewer #1: No

Reviewer #2: No

---

## [Author Response · Author response to Decision Letter 1]

1 Feb 2024

Rebuttal letter (PONE-D-22-31934)

Dear editorial team members, dear reviewers,

We very much appreciate the thorough, thoughtful, and constructive reviews we received for the manuscript "Understanding the effectiveness of a low-cost food package for ensuring food security at the household level during the COVID-19 pandemic: Difference-in-differences analyses of a community-based quasi-experimental trial". We also greatly appreciate the invitation to resubmit the revised manuscript. Below, we describe in detail how we have addressed each of the points raised by each of the Referees.

Thank you very much.

Corresponding author.

Reviewer #1: Thanks very much for the opportunity to review this interesting article. The development of a low-cost food basket is a much-needed intervention in the face of COVID-19 and other global threats, and evaluation of such an effort is critical. While factors contributing to the creation of the intervention is shared, more detail as to the targeting and overall approach would be valuable.

Response: Thank you so much for your comment. We have updated the methodology section accordingly.

The methodological details are very strong in some areas but omits key information. For example, there is not a clear picture of how control and intervention groups were sorted. This is particularly glaring because they are unbalanced in the outcome variable of interest at baseline. It’s possible that differences in findings thus are attributable to their initial differences in starting points, as opposed to the intervention. This point is not discussed and should be identified in the text as a key shortcoming of this study.

Response: Thank you so much for your suggestions. We have added details information about the participants selection in the methodology section (selection of the study participants). Also mentioned the possible shortcomings in the limitations section.

Additional comments are included as follows:

Check presentation of the findings in the abstract carefully, will be important to address methods used more clearly.

Response: Thank you for the comment. We have made the changes accordingly in the methodology section of abstract and main text as well.

Check for spelling errors in the abstract (eg, improv for impove).

Response: Thank you for the suggestion. We have Checked and corrected the spelling.

The writing throughout is organized into very long paragraphs. It would be more digestible if the paragraphs were more targeted.

Response: Noted. We have re-articulated the paragraph throughout the manuscript.

Introduction: There has been a lot of food insecurity data about the pandemic available at this point; studies in Britain are not the most relevant to the Bangladesh context. More global and Bangladeshi (if available) studies would be better context.

Response: Thank you so much for your valuable suggestion. We have incorporated information from some other studies from the similar context.

Orienting the introduction on acute food insecurity in disasters and the ways such a food package might have broader relevance to other scenarios (even beyond epidemic/pandemic) would be a valuable approach.

Response: Thank you for your suggestion. We have highlighted this issue in introduction section.

4 adults and 1 child is an odd target for the intervention given that’s not likely the average structure of households. How and why was this selected?

Response: Thank you for the queries. Prior to selecting the HH for intervention and control group, we conducted a rapid census in the selected geographic location. This data indicated that the median of adult members (more than 5 years old being counted as adult members) is 4, and the median of children is 1. This data also aligned with the nationally representative statistics in Bangladesh. The Bangladesh National Household Income and Expenditure survey indicated that the size of the average HH in Bangladesh is 4.2.

This was paired with extensive outcome measures for children, but if the intervention is only targeted to one child in the household how was this addressed?

Response: Thank you so much for bring this issue into light. Our household census data indicated that only a few (2 out of 157) households had more than one child at the time of the census. For this reason, we have designed the food package for one child and didn’t modify the food package for the additional children.

The study design is not adequately described. How were households determined to be intervention or control households? Was this randomized?

Response: We have added a section titled “selection of the study participants” where participant selection process is described in detail. This section indicated that participants had missed any meal during last month due to not having food in the household, have lost their livelihood due to the pandemic, had a monthly income less than or equal to 15000 BDT and were willing to provide their consent to participate were included into the study.

Selection criteria of food insecurity is included, but then households that are food secure are in the sample.

Response: Thank you for bringing up the issue. During household listing, we asked some specific questions to determine if households had missed any meals in the last month due to lack of food, lost their livelihood due to the pandemic, or had a monthly income that met the pre-defined criteria mentioned in the participant selection section of the manuscript. If they met these requirements, they were included in the sample frame for both arms of the study. It's important to note that the food security status of participating households was assessed with the HFIAS tool after they were enrolled in the study.

Substantial discussion of the uneven sample is needed – far more individuals were in the food insecure category for the intervention group, allowing for a great potential impact if the intervention is most effective for the severely food insecure. The limitations fail to discuss this.

Response: Thank you so much for bring the issue in light. We have mentioned this issue in the limitations section.

Reviewer #2: Editing is necessary. Abstract: More information on selecting intervention and control group must be presented.

Response: Thank you so much for the valuable suggestion. We have added the information in the abstract section.

Introduction: There are no literature on the effectiveness of food baskets to improve food insecurity in Bangladesh or other countries.

Response: Thank you for raising the issue. We already had some information in the discussion section. However, according to your suggestion, we have added some more information in the introduction and discussion section.

Methods: Considerations for developing food package needs references.

Response: Thank you for the query. We have discussed the whole procedure under the intervention section and added reference (ref 24).

Which criteria was used to select the control group?

Response: We have added the participant selection section in the main text and selection of control group has been discussed in detail in this section.

How many months did the intervention last?

Response: We provided the intervention for three months Thank you so much for the comment. We added this information is in the intervention section of the main text.

We used the Household Food Insecurity Access Scale to calculate the Household Food Insecurity Access Prevalence (HFIAP) based on their responses on the level of access to food and according to their score (Ref???).

Response: Thank you for addressing the issue. We have updated the reference.

Repeated measure ANOVA is recommended to detect the differences between intervention and control group adjusted for other variables.

Response: Thank you so much for the comment. We applied difference-in-differences analysis to measure the effect of the intervention as the differences between intervention and control group that was adjusted for other variables.

Results: The text of the findings is a repetition of the numbers in Tables and figures. Only significant differences are important to mention.

Response: Thank you so much. We have reviewed the results section and edited accordingly.

Figures 1 and 2 were not accessible.

Response: Thank you so much. We have updated the figures.

Discussion: Some discussions are unrelated to the results: “We found similar coping strategies in urban and rural areas in the baseline data. Some households received different types of support from various sources, but no systematic relief or safety net program was implemented in those areas.”

Response: Thank you so much. We have updated the text.

Comparisons of effectiveness with other studies have not been done.

Response: Thank you for your feedback. Unfortunately, we didn’t find any relevant study to compare during the manuscript preparation. That’s why we couldn’t compare the effectiveness of our intervention with other studies.

Acknowledgement: The authors would like to thank all the participants and their parents…???

Response: Thank you so much for addressing the issue. We have corrected the typo.

---

## [Decision Letter · Decision Letter 1]

2 Oct 2024

PONE-D-22-31934R1Understanding the effectiveness of a low-cost food package for ensuring food security at the household level during the COVID-19 pandemic: Difference-in-differences analyses of a community-based quasi-experimental trialPLOS ONE

Dear Dr. Das,

Thank you for submitting your manuscript to PLOS ONE. After careful consideration, we feel that it has merit but does not fully meet PLOS ONE’s publication criteria as it currently stands. Therefore, we invite you to submit a revised version of the manuscript that addresses the points raised during the review process.

We look forward to receiving your revised manuscript.

Kind regards,

Noor Raihani Zainol

Academic Editor

PLOS ONE

**Additional Editor Comments:**

It is a valuable study that can be made more useful by considering a few points.

Title: Since the project's findings are limited to a country, it is better to mention Bangladesh in the title.

Abstract: How was the food security status was measured?

Introduction: Previous studies on the provision of food aids by international organizations in Bangladesh such as WFP are not mentioned.

Methods: The amount of macro- and micronutrients provided by the package is not presented.

Eligibility criteria for low-cost food package is not clear.

There is no reference in the following sentence:

“We used the Household Food Insecurity Access Scale to calculate the Household Food Insecurity Access Prevalence (HFIAP) based on their responses on the level of access to food and according to their score (Ref).”

Measuring socio-economic variables were mot mentioned.

Results: The figures in the tables are repeated in the text.

Discussion: The main outcome variable of food security is different in two intervention and control groups. Researchers need to discuss whether it influenced the study results and whether there was a need for matching at the beginning of the study.

No findings were presented about the following discussion:

“We found similar coping strategies in urban and rural areas in the baseline data. Some households received different types of support from various sources,…”

Reviewers' comments:

Reviewer's Responses to Questions

**Comments to the Author**

1. If the authors have adequately addressed your comments raised in a previous round of review and you feel that this manuscript is now acceptable for publication, you may indicate that here to bypass the “Comments to the Author” section, enter your conflict of interest statement in the “Confidential to Editor” section, and submit your "Accept" recommendation.

Reviewer #2: All comments have been addressed

2. Is the manuscript technically sound, and do the data support the conclusions?

Reviewer #2: Yes

3. Has the statistical analysis been performed appropriately and rigorously? 

Reviewer #2: Yes

4. Have the authors made all data underlying the findings in their manuscript fully available?

Reviewer #2: Yes

5. Is the manuscript presented in an intelligible fashion and written in standard English?

Reviewer #2: Yes

6. Review Comments to the Author

Reviewer #2: It is a valuable study that can be made more useful by considering a few points.

Title: Since the project's findings are limited to a country, it is better to mention Bangladesh in the title.

Abstract: How was the food security status was measured?

Introduction: Previous studies on the provision of food aids by international organizations in Bangladesh such as WFP are not mentioned.

Methods: The amount of macro- and micronutrients provided by the package is not presented.

Eligibility criteria for low-cost food package is not clear.

There is no reference in the following sentence:

“We used the Household Food Insecurity Access Scale to calculate the Household Food Insecurity Access Prevalence (HFIAP) based on their responses on the level of access to food and according to their score (Ref).”

Measuring socio-economic variables were mot mentioned.

Results: The figures in the tables are repeated in the text.

Discussion: The main outcome variable of food security is different in two intervention and control groups. Researchers need to discuss whether it influenced the study results and whether there was a need for matching at the beginning of the study.

No findings were presented about the following discussion:

“We found similar coping strategies in urban and rural areas in the baseline data. Some households received different types of support from various sources,…”

7. PLOS authors have the option to publish the peer review history of their article (what does this mean? ). If published, this will include your full peer review and any attached files.

**Do you want your identity to be public for this peer review?** For information about this choice, including consent withdrawal, please see our Privacy Policy .

Reviewer #2: No

---

## [Author Response · Author response to Decision Letter 2]

13 May 2025

Rebuttal letter (PONE-D-22-31934R1)

Dear editorial team members, dear reviewers,

We very much appreciate the thorough, thoughtful, and constructive reviews we received for the manuscript "Understanding the effectiveness of a low-cost food package for ensuring food security at the household level during the COVID-19 pandemic: Difference-in-differences analyses of a community-based quasi-experimental trial in Bangladesh". We also greatly appreciate the invitation to resubmit the revised manuscript. Below, we describe in detail how we have addressed each of the points raised by the editors.

Thank you very much.

Corresponding author.

Additional Editor Comments:

1. It is a valuable study that can be made more useful by considering a few points.

Title: Since the project's findings are limited to a country, it is better to mention Bangladesh in the title.

Response. Thank you so much for the valuable suggestion. We have added it accordingly.

2. Abstract: How was the food security status was measured?

Response. We used Household Food Insecurity Access Scale and mentioned it in the Abstract.

3. Introduction: Previous studies on the provision of food aids by international organizations in Bangladesh such as WFP are not mentioned.

Response. Thanks for your valuable suggestions. we have added the information in the introduction section.

4. Methods: The amount of macro- and micronutrients provided by the package is not presented.

Response. Following the dietary guidelines for Bangladesh, we didn’t provide any separate micro and macronutrients in the package. However, Nutritional value of the food package were calculated using the food composition table. Table 1 details the energy content of the food package.

5. Eligibility criteria for low-cost food package is not clear.

Response. Thanks for highlighting the issue. We have discussed the food package selection process in the intervention section. We have considered the World Health Organization recommended energy requirement for an adult person is 2420 Kcal per day, and for a child is 1290 Kcal for finalising the food amount and technical expert group report of Bangladesh government to finalise the food items.

6. There is no reference in the following sentence: “We used the Household Food Insecurity Access Scale to calculate the Household Food Insecurity Access Prevalence (HFIAP) based on their responses on the level of access to food and according to their score (Ref).”

Response. We have added the reference accordingly. Please see reference no 32. Thanks for pointing this out.

7. Measuring socio-economic variables were not mentioned.

Response. Thanks for the suggestion. We have added accordingly.

8. Results: The figures in the tables are repeated in the text.

Response. Thanks for your comment on it. Following you comment we have re-write the section accordingly.

9. Discussion: The main outcome variable of food security is different in two intervention and control groups. Researchers need to discuss whether it influenced the study results and whether there was a need for matching at the beginning of the study.

Response. Thanks for mentioning the issue. We have discussed the issue accordingly in the limitation section

10. No findings were presented about the following discussion:m“We found similar coping strategies in urban and rural areas in the baseline data. Some households received different types of support from various sources,…”

Response. Thanks for the comment. We have added the findings in the result section.

---

## [Decision Letter · Decision Letter 2]

23 Jul 2025

PONE-D-22-31934R2Understanding effectiveness of a low-cost food package for ensuring food security at the household level: difference-in-differences analyses of a community-based quasi-experimental trial in BangladeshPLOS ONE

Dear Dr. Subashish,

Thank you for submitting your manuscript to PLOS ONE. After careful consideration, we feel that it has merit but does not fully meet PLOS ONE’s publication criteria as it currently stands. Therefore, we invite you to submit a revised version of the manuscript that addresses the points raised during the review process.

**ACADEMIC EDITOR:** 

**Title** : Since the study is based solely in Bangladesh, the title should reflect this geographical scope.**Abstract** : Please specify how food security status was measured.**Introduction** : Include a brief overview of previous food aid interventions in Bangladesh, particularly by international organizations such as the WFP, to contextualize your study.**Methods** :Provide details on the **macro- and micronutrient content** of the food packages.Clarify the **eligibility criteria** used to determine who received the food packages.Provide a reference for the use of the **Household Food Insecurity Access Scale (HFIAS)** , which is currently missing.Include a description of how **socio-economic variables** were measured and analyzed.**Results** : Avoid repeating detailed figures in both the tables and the narrative text.**Discussion** :Address the potential impact of baseline differences in food security between intervention and control groups, and whether matching or further statistical control was needed.Clarify the source of the statement about coping strategies and support received by households, and ensure relevant findings are presented to support the discussion.These revisions will help strengthen the clarity, replicability, and policy relevance of your findings. Please ensure your revision includes a **point-by-point response** to each issue raised and highlight any substantial changes in the manuscript.Indicate which changes you require for acceptance versus which changes you recommendAddress any conflicts between the reviews so that it's clear which advice the authors should followProvide specific feedback from your evaluation of the manuscript

We look forward to receiving your revised manuscript.

Kind regards,

Noor Raihani Zainol

Academic Editor

PLOS ONE

Journal Requirements:

Additional Editor Comments:

Title: Since the study is based solely in Bangladesh, the title should reflect this geographical scope.

Abstract: Please specify how food security status was measured.

Introduction: Include a brief overview of previous food aid interventions in Bangladesh, particularly by international organizations such as the WFP, to contextualize your study.

Methods:

Provide details on the macro- and micronutrient content of the food packages.

Clarify the eligibility criteria used to determine who received the food packages.

Provide a reference for the use of the Household Food Insecurity Access Scale (HFIAS), which is currently missing.

Include a description of how socio-economic variables were measured and analyzed.

Results: Avoid repeating detailed figures in both the tables and the narrative text.

Discussion:

Address the potential impact of baseline differences in food security between intervention and control groups, and whether matching or further statistical control was needed.

Clarify the source of the statement about coping strategies and support received by households, and ensure relevant findings are presented to support the discussion.

These revisions will help strengthen the clarity, replicability, and policy relevance of your findings. Please ensure your revision includes a point-by-point response to each issue raised and highlight any substantial changes in the manuscript.

Reviewers' comments:

Reviewer's Responses to Questions

**Comments to the Author**

1. If the authors have adequately addressed your comments raised in a previous round of review and you feel that this manuscript is now acceptable for publication, you may indicate that here to bypass the “Comments to the Author” section, enter your conflict of interest statement in the “Confidential to Editor” section, and submit your "Accept" recommendation.

Reviewer #3: (No Response)

2. Is the manuscript technically sound, and do the data support the conclusions?

Reviewer #3: Partly

3. Has the statistical analysis been performed appropriately and rigorously? 

Reviewer #3: Yes

4. Have the authors made all data underlying the findings in their manuscript fully available?

Reviewer #3: Yes

5. Is the manuscript presented in an intelligible fashion and written in standard English?

Reviewer #3: No

6. Review Comments to the Author

Reviewer #3: Overall comment:

Thanks for sharing the very important manuscript which focused on understanding effectiveness of a low-cost food package for ensuring food security at the household level. The manuscript is not written in standard English, therefore requires editing by a native English speaker which is mandatory before any decision being made.

Minor comments:

Title page

• Title should contain the timing of COVID-19

• Line 5: Mention (icddr,b) after Bangladesh before Dhaka

• Line 13: Same as line 5

• Line 14: 1206 will be 1212

Abstract

• Revise the whole abstract keeping within 300 words and using standard English

Introduction

• In the first paragraph, the word “worldwide” used several times. Kindly use synonyms.

• Line 61-62: Fonts are different. Kindly make it same throughout the manuscript.

Methods

• Line 86: km2 will be km2

• Line 97: Add (,) after participate

• Line 99: households will be HH

• Line 145: Replace field research assistants by FRAs

• Line 149: Delete “Food Frequency Questionnaire”

• Line 158: Report will be reported

• Line 160-161: Replace Household Food Insecurity Access Scale by HFIAS

• Line 170: under five will be under-five

Results

• Line 210: illustrates will be illustrated

• Line 215: indicate will be indicated

• Line 221: indicate will be indicated

Discussion

• Line 234: show will be showed

• Line 238: shot term will be short-term

• Line 240: show will be showed

• Line 268: Behavioral change communication will be Behavioral Change Communication

Acknowledgement

• Check the list of core donors

Additional comment: All the tables and figures need to be uploaded following journal style

7. PLOS authors have the option to publish the peer review history of their article (what does this mean? ). If published, this will include your full peer review and any attached files.

**Do you want your identity to be public for this peer review?** For information about this choice, including consent withdrawal, please see our Privacy Policy .

Reviewer #3: No

---

## [Author Response · Author response to Decision Letter 3]

5 Dec 2025

1. Title: Since the study is based solely in Bangladesh, the title should reflect this geographical scope.

Response: Thank you so much for the valuable suggestion. We have added it accordingly. Please see line 3 in the manuscript file.

2. Abstract: Please specify how food security status was measured.

Response: We used Household Food Insecurity Access Scale and mentioned it in the Abstract. Please see line 27-28 in the manuscript file.

3. Introduction: Include a brief overview of previous food aid interventions in Bangladesh, particularly by international organizations such as the WFP, to contextualize your study.

Response: - thanks for your valuable suggestions. we have added the information in the introduction section. Please see line: 74 to 76 where the information is included.

4. Methods:

o Provide details on the macro- and micronutrient content of the food packages.

Response: following the guideline we didn’t provide any separate micro and macronutrients in the package. However, Nutritional value of the food package was calculated using the food composition table. Please see table 1 for details

o Clarify the eligibility criteria used to determine who received the food packages.: Thank you for the comment. We discussed the eligibility criteria in the selection of the study participant’s section. Please see line 91-95 in the manuscript.

o Provide a reference for the use of the Household Food Insecurity Access Scale (HFIAS), which is currently missing.: Thanks for sharing the concern. We have added accordingly. Please see reference no 32.

o Include a description of how socio-economic variables were measured and analyzed.: Thanks for the suggestion. We have added accordingly in the data analysis section

5. Results: Avoid repeating detailed figures in both the tables and the narrative text.: Thanks for your comment on it. Following you comment we have re-write the section accordingly.

6. Discussion:

o Address the potential impact of baseline differences in food security between intervention and control groups, and whether matching or further statistical control was needed.; Thanks for mentioning the issue. We have discussed the issue accordingly in the limitation section. Please see line 284-287 for the updated writeup.

o Clarify the source of the statement about coping strategies and support received by households, and ensure relevant findings are presented to support the discussion.: - Thanks for the comment. Coping strategies in the result section are discussed in between 208-210. Also, relevant findings are presented in between lines 241-244 in the discussion section.

Title page

• Title should contain the timing of COVID-19

Response: Added in the title

• Line 5: Mention (icddr,b) after Bangladesh before Dhaka

Response: Made the change

• Line 13: Same as line 5

Response: Made the change

• Line 14: 1206 will be 1212

Response: Made the change

Abstract

• Revise the whole abstract keeping within 300 words and using standard English

Response: we have revised the whole abstract, Shorten it within 300 words and made the changes in writing pattern.

Introduction

• In the first paragraph, the word “worldwide” used several times. Kindly use synonyms.

Response: Made the change in the first paragraph

• Line 61-62: Fonts are different. Kindly make it same throughout the manuscript.

All are changed into Times new Roman (11 pt)

Methods

• Line 86: km2 will be km2

Corrected

• Line 97: Add (,) after participate

Added

• Line 99: households will be HH

Corrected

• Line 145: Replace field research assistants by FRAs

Corrected

• Line 149: Delete “Food Frequency Questionnaire”

Corrected

• Line 158: Report will be reported

• Line 160-161: Replace Household Food Insecurity Access Scale by HFIAS

Corrected

• Line 170: under five will be under-five

Corrected

Results

• Line 210: illustrates will be illustrated

Corrected

• Line 215: indicate will be indicated

Corrected

• Line 221: indicate will be indicated

Corrected

Discussion

• Line 234: show will be showed

Corrected

• Line 238: shot term will be short-term

Corrected

• Line 240: show will be showed

Corrected

• Line 268: Behavioral change communication will be Behavioral Change Communication

Changed accordingly

Acknowledgement

• Check the list of core donors

Updated accordingly

---

## [Decision Letter · Decision Letter 3]

24 Feb 2026

Understanding effectiveness of a low-cost food package for ensuring food security during the COVID-19 at the household level: difference-in-differences analyses of a quasi-experimental trial in Bangladesh

PONE-D-22-31934R3

Dear Dr. Das,

We’re pleased to inform you that your manuscript has been judged scientifically suitable for publication and will be formally accepted for publication once it meets all outstanding technical requirements.

Kind regards,

Neetu Choudhary, PhD

Academic Editor

PLOS One

Additional Editor Comments (optional):

Reviewers' comments:

Reviewer's Responses to Questions

**Comments to the Author**

1. If the authors have adequately addressed your comments raised in a previous round of review and you feel that this manuscript is now acceptable for publication, you may indicate that here to bypass the “Comments to the Author” section, enter your conflict of interest statement in the “Confidential to Editor” section, and submit your "Accept" recommendation.

Reviewer #3: All comments have been addressed

Reviewer #4: All comments have been addressed

2. Is the manuscript technically sound, and do the data support the conclusions?

Reviewer #3: Yes

Reviewer #4: Yes

3. Has the statistical analysis been performed appropriately and rigorously? 

Reviewer #3: Yes

Reviewer #4: Yes

4. Have the authors made all data underlying the findings in their manuscript fully available?

Reviewer #3: No

Reviewer #4: Yes

5. Is the manuscript presented in an intelligible fashion and written in standard English?

Reviewer #3: Yes

Reviewer #4: No

6. Review Comments to the Author

Reviewer #3: Authors have addressed all the comments provided; thus, I recommend acceptance of the manuscript and publication to the journal.

Reviewer #4: Authors must improve their English in accordance with standard international scientific article writing conventions.

7. PLOS authors have the option to publish the peer review history of their article (what does this mean? ). If published, this will include your full peer review and any attached files.

**Do you want your identity to be public for this peer review?** For information about this choice, including consent withdrawal, please see our Privacy Policy .

Reviewer #3: No

Reviewer #4: No

---

## [Editor Report · Acceptance letter]

PONE-D-22-31934R3

PLOS One

Dear Dr. Das,

I'm pleased to inform you that your manuscript has been deemed suitable for publication in PLOS One. Congratulations! Your manuscript is now being handed over to our production team.

Kind regards,

on behalf of

Dr. Neetu Choudhary

Academic Editor

PLOS One